# The Analysis of Company Growth Determinants Based on Financial Statements of the European Companies

Bojana Vuković , Kristina Peštović *, Vera Mirović, Dejan Jakšić and Sunčica Milutinović

Faculty of Economics in Subotica, University of Novi Sad, 24000 Subotica, Serbia;
bojana.vukovic@ef.uns.ac.rs (B.V.); vera.mirovic@ef.uns.ac.rs (V.M.); dejan.jaksic@ef.uns.ac.rs (D.J.);
suncica.milutinovic@ef.uns.ac.rs (S.M.)
* Correspondence: kristina.pestovic@ef.uns.ac.rs; Tel.: +381-63-582-776

**Abstract:** The primary aim of this paper is to determine the indicators that have an influence on the company growth in the field of agriculture, forestry, and fisheries during the six-year period (2014–2019). This sector is very important for sustainable development, bearing in mind the need to preserve natural resources, i.e., land, water, plant, and animal resources. Sustainable development of this sector is of satisfactory technical-technological development, economically sustainable, and socially acceptable. The sample consists of 1333 observations of active companies on the European market. Multiple regression analysis was used in order to thoroughly analyze the variables of growth. The obtained results showed that company size has a negative impact on growth, while return on assets and leverage have a positive impact on growth. The impact of these variables was statistically significant. Along with the influence of observed determinants based on data from financial statements, the future growth and development of companies in this sector will certainly depend on the volume of investments, pricing policy, credit and natural conditions, agricultural policy measures, and adequate institutional support through the provision of financial support and encouragement of exports of products. Institutional incentives for more intensive integration of the agriculture, forestry, and fisheries sector are aimed at achieving the concept of integrated sustainable development.

**Keywords:** growth; agriculture; forestry and fisheries; European companies



## 1. Introduction

The growth of each company presupposes an increase in the size of the company in a certain period of time. Businesses can expand while remaining within their regular range of products or services, or they can expand their range to provide diversity. Business diversity affects more stable business conditions and leads to a reduction in business risk. The extent to which companies will diversify is determined by their size and financial strength. Recognizing the importance of growth is aimed at creating conditions that will lead to economic prosperity. Growth affects the increase of value and strengthening of the company, as well as the increase in economic activity. Fast-growing companies contribute to the growth of the world economy.

Potential growth shows how a company develops the ability or capacity for growth and development in the near future. Growth rate indicates heterogeneity that exists between companies and is difficult to predict [1]. A good growth rate is achieved in accordance with economic development and there is no problem from the aspect of survival and development of companies. There is no constraint from the aspect of company growth if resources are used efficiently. The neoclassical theory assumes that companies should achieve optimal size in order to maximize profits. So, there is an optimal size of the company that all companies are approaching. When the optimal size is established, it is assumed that companies do not grow above the optimal size. Company growth is based

on all the benefits that managers attribute to the size of the company. In circumstances when there are different preferences of managers and shareholders, managers who act in the interest of shareholders should make the appropriate choice between maximizing profits on the one hand or pursuing their interests in the direction of maximizing growth, on the other. Efficient companies usually reinvest their profits in order to grow, and on that basis, we can conclude that more efficient companies achieve higher growth rates. The annual rate of company growth is conditioned by the accuracy with which managers can predict product prices. Understanding potential growth by the way of annual growth rate is paramount for stock market investors.

Internal growth is a key measure of company success, so it is very often possible to equate growth with success. There are two primary schools that identify key factors influencing growth. The first group of factors starts from the influence of the size and age on growth, while the second group of factors is based on variables such as strategy, organization and characteristics of the owner or manager of the company [2]. Potential growth in this paper will be viewed as the estimated growth of the company based on financial parameters that could lead to the growth. The responsibility of managers is reflected in the detailed consideration of the impact of financial parameters in order to achieve growth. The assessment of the potential growth of the company in this paper is performed on the basis of financial statements. The paper should indicate whether there are financial constraints on the company growth and what are the most significant indicators to consider in the context of assessing the company growth.

The obtained results are directed towards the managers of the companies in the field of agriculture, forestry, and fisheries in order to ensure the growth of business efficiency. This paper was mostly motivated by the insufficient scope of research on the policy of growth determinants of companies. To our knowledge, there is no research on the policy of company growth on a sample of European companies operating in the agriculture, forestry, and fisheries sectors. Likewise, previous research on the determinants of company growth has focused on industries other than agriculture, forestry, and fisheries [3–7].

Furthermore, previous research did not include companies in this sector operating in the European market. The growth of companies in the European market was researched only by Malinić et al. at the level of ten central and eastern European countries. Observing country-specific and company-specific variables, Malinić et al. reexamined the company growth determinants during the crisis and post-crisis period [8].

The aim of this paper was to build up a model that pointed out the indicators of growth policy of the companies in the field of agriculture, forestry, and fisheries in European countries. Companies in this field face high production costs, slow asset turnover, susceptibility to the seasonal nature of production, and high production risk which affects the lower volume of capital investment. The natural factors have a great influence: there is a natural character of production and the duration of production differs from the duration of direct work. Identifying the determinants that lead to the growth of companies will ensure the development of this sector, providing an incentive to reduce imports and increase exports, which will certainly have a positive impact on the economic growth of European countries. Research conducted by Růčková and Škuláňová showed that companies in agriculture, forestry, and fisheries industries in seven countries of central and eastern Europe primarily decided to finance from their own sources in a period of economic growth when profit growth most often occurs [9].

There is a potential of the agriculture, forestry, and fisheries sector as a fast-growing sector on the European market so that efficient management of this sector provides food on the world market. Identification of key growth variables of companies in this sector is aimed at increasing the value and efficiency of agricultural production, as well as the volume of necessary investments for the development of the processing industry. Government support is needed for satisfactory company growth through the policy of price and quality management, taking into account the control of prices of essential products, stocks, and raw materials. In this way, the formation of monopolies and market manipulation would

be prevented, which leads to the creation of added value for the agricultural product. Providing sufficient and secure inputs to the agricultural, forestry, and fisheries industry will also ensure economic and socio-political stability. The results also provide guidance on how to become a high-growth company and also have implications for the policies of high-growth companies in the agriculture, forestry, and fisheries industry.

The paper should answer the following two questions:

- Based on data from financial statements, what are the internal determinants of company performance that have an impact on the growth policy of the companies in the field of agriculture, forestry, and fisheries in European countries?
- What is the relationship between the internal determinants and company growth policy in the field of agriculture, forestry, and fisheries in European countries?

The analysis of the determinants of company growth in the field of agriculture observed from the aspect of sustainable development takes into account the requirements aimed at creating economically viable and environmentally friendly agricultural production. In the context of this goal, the observed agricultural companies should take advantage of natural preconditions and provide growth in order to achieve a satisfactory level of competitiveness and ensure performance in markets outside the European one. Bearing in mind that the specifics of forestry and fishery as activities are conditioned by naturally determined quantity and quality of resources, when analyzing the business of these companies and identifying determinants that affect growth, it is necessary to keep in mind sustainable development goals. They are reflected in sustainable forest management and forestry improvement, improving fisheries through appropriate management of fish resources, and increasing the economic effects that can be achieved through the rational use of forest and fisheries resources.

The paper analyzed the impact of five independent variables (the size of the company, current ratio, return on total assets, return on equity, leverage) on the company growth, measured by the sales growth ratio in the industry of agriculture, forestry, and fisheries in European countries. The paper is structured in the following way: we first start with Theoretical Background and Hypothesis Development, then continue with the Methodology. After presentation of Results and Discussion, the Conclusion summarizes the limitations and recommendations for future research.

## 2. Theoretical Background and Hypothesis Development

This research is aimed at determining the factors that influence company growth. There are many ways in which growth can be measured. It can be asset growth, profit growth, turnover growth, or growth in operating revenues. In this paper, growth is represented as a change in sales. The sales growth rate is the rate of increase in sales revenue of a company over a certain period of time. Kachlami and Yazdanfar summarize the key benefits of using sales as a growth indicator as follows: sales growth rate and the number of employees are the most commonly used determinants of growth in research studies and some other growth determinants have certain disadvantages so that they can only be used in specific circumstances [10]. For example, the value of total assets is related to the industry capital intensity and is sensitive to changes in time. Alese and Alimi believed that sales growth data are easily accessible, applicable to all kinds of companies, and are not subject to changes in terms of capital intensity and integration degree [11]. Sampagnaro confirmed that sales growth is an indicator which is easy to determine, regardless of the industry in which the companies operate, and which does not react to changes in the horizontal and vertical integration of production [12]. In addition, it is an indicator that entrepreneurs opt for and a variable that conditions other variables. Relying on previous empirical research in this area, the impact of the following variables on sales growth was measured: the size of the company, current ratio, return on assets, leverage, and return on equity.

There are several theoretical views on the creation and growth of companies. The most famous among them is Gibrat's law which assumes that every company, regardless of size, strives to achieve a certain growth rate. Gibrat's law also defines the rules of

proportional growth of a company, starting from the fact that the proportional growth rate of a company is independent of its size [5]. Some companies grow, develop, and record success, while others do not develop and cease to exist, leaving the market. Small and medium-sized companies often do not rely on the mentioned Gibrat's law and establish strategic growth guidelines in order to achieve the minimum level of efficiency necessary for market survival. Consequently, small- and medium-sized companies are expected to have higher growth than large companies, confirming the negative relationship between size and growth. On the other hand, large companies achieve efficiency at a certain level, so that they have already reached the minimum efficiency rate required for market survival and their growth can be completely independent of size.

Research conducted by Mateev and Anastasov consisted of 560 fast-growing small- and medium-sized companies from six transition economies and showed that firm size to a large extent has explained the growth in small and medium-sized companies in central and eastern European countries [2]. As expected, the company size has a statistically significant positive impact on company growth. Thus, the company size has an impact on increasing sales revenues. Kachlami and Yazdanfar analyzed the financial growth determinants of companies in Sweden, concluding that larger companies achieve higher growth rates and confirmed the positive relationship between size and growth [10]. Larger companies are more inclined than smaller ones to diversify their activities and expand into larger markets. Niskanen and Niskanen pointed out that the growth of small and micro companies in Finland increases with company size to a certain level and then starts to decline. Interestingly, the impact is statistically significant only in the case of companies with less than ten employees. By observing manufacturing and nonmanufacturing companies, they rejected the assumption of Gibrat's law only for manufacturing companies [13]. On the other hand, research conducted by Gill and Mathur analyzed the growth determinants of 164 Canadian companies listed on the Toronto Stock Exchange in manufacturing and service industries in the time period from 2008 to 2010 [4]. The results showed that there was a negative relationship between potential company growth and firm size. Becchetti and Trovato also presented an empirical analysis of the determinants of growth for a sample of Italian small and medium-sized companies which included companies between 10 and 50 employees [14]. Factors that significantly negatively affect company growth are size and age. Carvalho et al. analyzed a sample of 182 small and medium-sized companies (SMEs) of the fitness industry in the time period from 2004 to 2009 [5]. The results showed that the smaller fitness SMEs companies in Portugal grew faster than larger ones, so the company size has a statistically significant negative impact on growth and Gibrat's law assumption is rejected. This relationship was also confirmed in research conducted by Sampagnaro [12]. On the other hand, Hermelo and Vassolo concluded that there is no statistically significant relationship between company growth and size, which is in line with Gibrat's law [15]. Analyzing 444 growth strategies of 74 Spanish hotel companies for the time period of 2001 to 2003, Claver et al. found that larger companies have a larger volume of diversified resources and most often opt for riskier growth strategies [16].

Bearing in mind all previous research and especially research conducted by Mishra and Soumya [6], Aggarwal [17], Liu and Hsu [18], we set the following research hypothesis:

**Hypothesis 1 (H1).** *The company size has a statistically significant positive impact on company growth.*

Current liquidity shows the company ability to finance current liabilities with available current assets [19]. The impact of current liquidity on company growth has been analyzed in a large number of empirical studies. By predicting the rapid growth of 21,182 Italian manufacturing SMEs in 2003–2007, Sampagnaro found that a higher value of current liquidity indicates a lower degree of probability that the company will achieve high growth and thus confirms that reinvestment in liquidity is actually associated with the growth of the company [12]. Mishra and Soumya emphasized that maintaining the optimal level of

liquidity of the company through adequate liquidity management has a positive impact on the business results and leads to company growth [6]. Research conducted by Megaravalli and Sampagnaro analyzed the growth predictors of 45,000 family business companies in Italy [20]. The results showed that the company growth potential increases with liquidity increase and that the growth of liquidity indicators leads to the growth of the company capabilities and the growth of the efficiency of working capital management. The same authors in analyzing the liquidity ratio as predictors of company growth on the sample of 1905 manufacturing companies of India showed that the liquidity indicators are in fact one of the key determinants of company growth, and that a more favorable liquidity position affects the company higher growth and reduces the possibility of default. On the other hand, Bashir et al. by using the panel data model showed that there is a negative but insignificant impact of the liquidity on the company growth in the food and textile sectors of Pakistan for the duration of 2013 to 2017 [7]. Voulgaris et al. also found a negative relationship between current liquidity and growth of 143 fast-growing companies from the manufacturing SMEs sector in Greece by panel data analysis [3]. These companies made savings in the use of current funds and borrowed from other sources such as banks and trade creditors. Analyzing the growth determinants of 250 Indian firms for the time period of 2004-05 to 2013-14, Aggarwal showed that companies with high growth do not have high volume of liquid assets, confirming the significant negative association between liquidity and company growth [17].

Based on all presented research and research conducted by Megaravalli and Sampagnaro [20], the following hypothesis was posed:

**Hypothesis 2 (H2).** *Current liquidity has a statistically significant positive impact on company growth.*

Profitability is a key prerequisite for the long-term success and survival of the company that expresses the degree of return on the company's engaged assets [19]. It can also indicate the company's ability to return on invested funds of shareholders or what percentage of net profit is made by the owners of capital [21]. Analyzing the relationship between profitability and growth, Goddard et al. found that the profit realized in the current period is the basis for future company growth, since profit is the main source of finance for the development of the company [22]. On the other hand, excessive growth in the current period can negatively affect profit in the coming period. Their research showed that there is no essential relationship between the growth and profitability of the company. Research conducted by Yoo and Kim showed that profitability in the current period can be stimulated by the high growth of the previous period in the conditions of a stable macroeconomic environment [23]. Voulgaris et al. showed that the higher profitability measured by return on invested assets leads to higher growth rates [3]. In certain circumstances there is no significant relationship between profitability and sales growth, given that some companies can operate with a high level of profitability even in circumstances where there is a decline in the growth rate because of an oligopolistic structure in the market or entry barriers. Bashir et al. found that profitability as a return on invested equity should be considered by management and policymakers as a driving factor for increasing growth in the textile and food sector of Pakistan [7]. Companies that achieve rapid growth operate more profitably. Additionally, when companies enter the market in large numbers and quickly, they become more profitable. Analyzing the determinants of company growth for 280 Taiwan manufacturing companies for the time period of 1991 to 2002, Liu and Hsu confirmed that return on total assets as a measure of a company's ability to generate its resources has a significantly positive impact on growth [15]. Niskanen and Niskanen presented research about the determinants of company growth in small and micro Finnish companies, the results of which showed that the increase in profitability leads to the growth of companies with less than 10 employees and companies that are categorized as nonmanufacturing [13].

Exploring the impact of profitability on the growth of manufacturing companies in the food and beverages sectors listed on the Indonesia Stock Exchange in the time period from 2010 to 2012, Khaldun and Muda concluded that return on assets and return on equity together affect the growth of manufacturing companies [21]. The impact of these indicators individually is not statistically significant. Lastly, Claver et al. found that lower profitability and higher liquidity of Spanish hotel companies means that these companies are opting for less risky growth strategies in order to increase efficiency and strengthen the competitive position [15].

Keeping in mind the previously outlined research and research results of Sampagnaro [12], Aggarwal [17], and Zekić-Sušac et al. [24], the authors formulated the following hypotheses:

**Hypothesis 3 (H3).** *Profitability measured by return on assets has a statistically significant positive impact on company growth.*

**Hypothesis 4 (H4).** *Profitability measured by return on equity has a statistically significant positive impact on company growth.*

Leverage is an indicator that shows the extent to which a company relies on other sources of financing in financing its investments [25]. It can be defined as an approximate indicator of the company's ability to finance from other sources of financing. Analyzing the relationship between leverage and growth, Carvalho et al. conclude that there is a statistically significant positive relationship between these two variables [5]. But in crisis business conditions, financing from other sources is not a positive growth factor for small and medium-sized fitness companies in Portugal due to high interest paid. Thus, interest paid is a restrictive growth determinant of the Portuguese fitness SMEs and has a negative impact on the survival of these companies, which leads to the conclusion that excessive funding from other sources would have a negative impact on the operations of these companies. Honjo and Harada, by using a panel data set on 6961 small and medium-sized companies in the Japanese manufacturing industry in the time from 1995 to 1999 fiscal year, concluded that companies that have certain growth opportunities could actually be funded on an increased scale from other sources of financing to expand their sales opportunities [26]. In that way, Cassia et al. analyzed the factors that differentiate between high-growth and low-growth European companies in order to define the growth policy of European companies in the future [27]. The results showed that the high-growth companies were characterized by a large volume of current and future investments, high leverage, and growth. On the other hand, research conducted by Sampagnaro was aimed at assessing whether and to what extent there is a certain inverse relationship between indebtedness and growth and a positive relationship between internal cash flows and company growth [12]. The conclusion is that financing from other sources to some extent negatively affects the company growth. The same conclusions were reached by Bashir et al. who found that there is a statistically significant negative relationship between leverage and growth which indicates that leverage reduces company growth in the Pakistan food and textile sectors [7]. Liu and Hsu pointed out that high leverage is associated with low company growth of 280 Taiwan manufacturing companies indicating that a potentially good financial structure of companies will ensure their growth [15]. Niskanen and Niskanen showed that limited liability companies borrow to high-risk levels when borrowing begins to hinder growth. The research also showed that leverage is a significant determinant of companies with more than 10 employees [13]. Simbana et al. confirmed the negative association between indebtedness and growth by researching a sample of 41,333 Ecuadorian companies from all economic sectors for the time period of 2000 to 2013. Results showed that indebtedness growth accompanied by a recession cycle affects the ability to generate sufficient income, so that financial constraints can lead to growth constraints [28]. Agarwal concluded that there is a negative relationship between leverage and company growth which means that lower

leverage leads to better company performance. The results of his research showed that this relationship is not statistically significant [17]. Finally, research conducted by Claver et al. showed that the higher indebtedness of Spanish hotel companies for the time period of 2001 to 2003 leads to the implementation of more profitable and risky growth strategies so that they can settle their financial obligations [15].

Sublimating the results of previous empirical studies, we set the following hypothesis:

**Hypothesis 5 (H5).** *Leverage has a statistically significant positive impact on company growth.*

### 3. Methodology

The survey covered 1333 observations of large and very large companies for the time period of 2014 to 2019. Companies are categorized as public limited companies, private limited companies, limited liability companies, joint-stock companies, and private joint-stock companies. The advantages of large companies from a growth perspective are reflected in taking advantage of economies of scale, achieving greater market power, diversify their activities, expand into larger markets, negligible takeover risk, and higher status. The sample was structured according to the code of activity and included companies operating in the sector of agriculture, forestry, and fisheries on the European market (Table 1). Financial statements of companies in this sector located in TP Catalyst database were the basis for research [29]. The sample includes the active companies from the following countries: Belgium, Bulgaria, Chez Republic, Denmark, Estonia, Greece, Hungary, Croatia, Lithuania, Latvia, Norway, Russia, Slovakia, and Ukraine (Table 2).

**Table 1.** Sample structure according to activity of company.

| Activity | Number of Observation | Structure % |
|---|---|---|
| Aquaculture Industry | 31 | 2.33% |
| Breeding animal and production | 628 | 47.11% |
| Crop production | 160 | 12.00% |
| Fruits and vegetable production | 89 | 6.68% |
| Oilseed and Grain Farming Industry | 321 | 24.08% |
| Mixed farming | 41 | 3.08% |
| Silviculture and other forestry activities | 63 | 4.73% |
| Grand Total | 1333 | 100.00% |

Source: Author's calculation.

**Table 2.** Sample structure according to country.

| Country | Number of Observation | Structure % |
|---|---|---|
| BE | 33 | 2.48% |
| BG | 24 | 1.80% |
| CZ | 19 | 1.43% |
| DK | 4 | 0.30% |
| ES | 12 | 0.90% |
| GR | 6 | 0.45% |
| HR | 24 | 1.80% |
| HU | 192 | 14.40% |
| LT | 12 | 0.90% |
| LV | 12 | 0.90% |
| NO | 37 | 2.78% |
| RS | 6 | 0.45% |
| SK | 80 | 6.00% |
| UA | 872 | 65.42% |
| Grand Total | 1333 | 100.00% |

Source: Author's calculation.

The sector of agriculture, forestry, and fisheries is especially important for sustainable economic development which aims to provide constant long-term economic growth which will achieve a certain level of economic efficiency, better use of natural resources, and improvement of quality of life. Key indicators for the agriculture, forestry, and fisheries sector in observed European countries are in Table 3.

**Table 3.** Key indicators for the agriculture, forestry, and fisheries sector in observed European countries.

| Country | Contribution of Agriculture to Gross Domestic Product in 2019 (%) | Value of Agricultural Industry Output in 2019 (EUR Million) | Employment in Agriculture in 2019 (Thousand Annual Work Units) | Forest and Other Wooded Land in 2020 (Thousand Hectares) | Gross Value Added (at Basic Prices) in Forestry in 2017 (EUR Million) | Persons Employed in Forestry and Logging in 2017 (Thousand Annual Work Units) | Total Catches (Major Fisheries Areas) in 2019 (Tonnes Live Weight) | Total Aquaculture Production in 2018 (Tonnes Live Weight) | Persons Employed in Fisheries and Aquaculture in 2018 (Thousand Annual Work Units) |
|---|---|---|---|---|---|---|---|---|---|
| Belgium | 0.5 | 8713 | 55.6 | 722 | 83 | 2.3 | 21,061 | 0 | 0.4 |
| Bulgaria | 2.7 | 4348 | 190.4 | 3917 | 233 | 12.0 | 10,269 | 10,758 | 1.6 |
| Czechia | 0.8 | 5498 | 102.0 | 2677 | 1200 | 21.7 | 0 | 21,750 | 1.5 |
| Denmark | 1.1 | 11,629 | 53.7 | 665 | 296 | 6.0 | 0 | 32,167 | 2.0 |
| Estonia | 1.0 | 998 | 18.9 | 2533 | 249 | 5.5 | 83,626 | 944 | 0.7 |
| Greece | 3.1 | 11,880 | 416.9 | 6537 | 66 | 9.3 | 82,232 | 132,413 | 20.6 |
| Croatia | 1.9 | 2423 | 176.4 | 2557 | 196 | 13.8 | 64,020 | 19,680 | 4.9 |
| Latvia | 1.6 | 1629 | 70.0 | 3519 | 393 | 17.2 | 0 | 828 | 1.5 |
| Lithuania | 1.8 | 3209 | 134.6 | 2263 | 212 | 13.2 | 100,691 | 3446 | 1.1 |
| Hungary | 2.2 | 8722 | 358.9 | 2253 | 249 | 20.6 | 0 | 17,900 | 1.4 |
| Slovakia | 0.5 | 2261 | 44.5 | 1946 | 426 | 19.7 | 0 | 2247 | 0.3 |

Source: Author's illustration according to Eurostat [30].

Analyzing key indicators presented in Table 3, we can conclude that agriculture in Greece has the highest contribution to the gross domestic product in 2019 (3.1%). On the other hand, the lowest share in the gross domestic product in 2019 has the agriculture of Belgium and Slovakia (0.5%). The highest value of agricultural industry output in 2019 has Greece with 11,880 million EUR. Estonia is a country with the lowest value of agricultural industry output of 998 million EUR in 2019. Observing the forestry sector, the results showed that Greece has the largest area under forests and other wooded land of 6537 thousand hectares in 2020. Denmark is a country with the smallest area under forests and other wooded land of 665 thousand hectares in 2020. Judging by the gross value added in forestry in 2017, Czechia ranks first with 1200 million EUR while Greece is in the last place (66). Bearing in mind the fishery sector, especially the value of total catches in major fisheries areas in 2019, Lithuania achieves the highest value of 100,691 tonnes live weight. Further, the highest value of total Aquaculture Production in 2018 has Greece with 132,413 tonnes live weight.

Analyzing the employment in the agriculture, forestry, and fisheries sector measured by thousand annual work units, we can conclude that the highest employment in the agricultural sector was in Greece in 2019. The lowest employment in the agricultural sector in 2019 is recorded in Estonia. The highest employment in forestry and logging in 2017 is achieved by Czechia, while the highest employment in fisheries and aquaculture in 2018 was in Greece. On the other hand, the worst results from the aspect of employment in forestry and logging in 2017 was Belgium, or Slovakia in fisheries and aquaculture in 2018. To sum up, the Greek market has the best potential for the development of the agriculture, forestry, and fisheries sector in the observed period.

Data were processed in the statistical program Stata 13. The sales growth rate was observed as a dependent variable. As independent variables were observed the size of the company, current ratio, return on total assets, return on equity, and leverage. Independent variables were selected as the most commonly used financial indicators influencing

company growth in previous empirical studies. Implemented indicators and the method of their calculation were presented in Table 4 relying on the research conducted by Voulgaris et al. [3]; Mateev and Anastasov [2]; Carvalho et al. [5]; Sampagnaro [12]; Khaldun and Muda [21]; Kachlami and Yazdanfar [10]; Zekić-Sušac et al. [24]; Megaravalli and Sampagnaro [31]; Megaravalli and Sampagnaro [20]; Mishra [22]; and Bashir et al. [7].

**Table 4.** Determinants that may impact growth based on data from financial statements.

| Indicators | Method of Calculation |
|---|---|
| Sales Growth Rate | (Sales of Current Period-Sales of Previous Period)/ Sales of Previous Period |
| Size | Log of Number of Employees |
| Liquidity | Current Assets/Current Liabilities |
| Profitability | ROA-Net Income/Total Assets |
| Profitability | ROE- Net Income/Equity |
| Leverage | Debt/Asset |

Source: Author's illustration.

Empirical analysis of observed variables consisted of descriptive statistics, correlation matrix, and multiple regression analysis. There were 1608 initial observations. The model consisted of 1333 observations after the elimination of missing or abnormal values. Based on previous research conducted by Aggarwal [15] and Khaldun and Muda [21] and in order to identify the main factors of the growth, the following multiple regression model was set:

$$Growth_{it} = \beta_0 + \beta_1 X_1 + \beta_2 X_2 + \beta_3 X_3 + \beta_4 X_4 + \beta_5 X_5 + \epsilon i$$

$Growth_{it}$—dependent variable;
$\beta_0$—model constant;
$\beta_i$—coefficiency of independent variables;
$X_1$—Size of the Company (independent variable)
$X_2$—Current Liquidity (independent variable)
$X_3$—Profitability measured by ROA (independent variable)
$X_4$—Profitability measured by ROE (independent variable)
$X_5$—Leverage (independent variable)
E—error with a normal distribution;
i—signify each company (i = 1, . . . , $N$);
t—signify the period of time (t = 1, . . . , t).

## 4. Results and Discussion

Descriptive statistics of dependent and independent variables of the analyzed model are presented in Table 5.

**Table 5.** Descriptive statistics.

| | N | Minimum | Maximum | Mean | Std. Deviation |
|---|---|---|---|---|---|
| Growth | 1333 | −0.9553 | 28.1232 | 0.2615 | 1.4885 |
| Size | 1333 | 0.6931 | 8.7327 | 5.0780 | 1.1696 |
| Current Ratio | 1333 | 0.1540 | 77.7180 | 4.3745 | 8.0818 |
| ROA | 1333 | −88.5530 | 99.9100 | 9.1021 | 15.6832 |
| ROE | 1333 | −225.5582 | 410.6464 | 18.5666 | 42.5192 |
| Leverage | 1333 | 0.0084 | 4.2228 | 0.4777 | 0.4482 |
| Valid N (listwise) | 1333 | | | | |

Source: Author's calculation.

The average value of the growth rate was 0.2615 with a discrepancy that varied from a minimum value of −0.9553 to a maximum value of 28.1232. There was a large discrepancy between companies from a growth perspective. The negative value of the growth rate of some companies indicates that some companies did not achieve sales growth in the observed period. The average value of company size was 5.0780 with no significant value dispersions. The average value of the current liquidity ratio was 4.3745, which is in accordance with the reference value (≥2). Analyzed companies had liquid business or 4.3745 higher assets to finance liabilities than the value of liabilities in the short-term. The discrepancy in the current ratio varied from a minimum value of 0.1540 to a maximum value of 77.7180, which indicates that in the observed sample there are companies that operate with an extremely high degree of current liquidity. The average rate of Return on Assets was 9.1021%. Bearing in mind that the reference value of ROA is ≥10%, it is obvious that the observed companies did not achieve the reference value from the aspect of acceptable rate of return on engaged assets. Judging by the significant value dispersion of ROA from −88.5530 to 99.9100, it can be concluded that there are companies that do not achieve growth, on the one hand, and companies with extremely high growth, on the other. The average rate of ROE was 18.5666% which is in accordance with the reference value. It indicates the profitable business of these companies. Return on Equity indicator also showed significant value dispersion, from −225.5582 to 410.6464. Therefore, the observed sample consisted of companies operating at a loss, but also of companies operating with an extremely high return on engaged equity in the observed period. The Debt to Asset indicator had an average value of 0.4777 which means that total assets were 48% financed from debts. Value dispersion of this indicator varies from 0.0084 to 4.2228 which means that in the observed sample there are companies with four times the value of debts in relation to the value of total assets.

Descriptive statistics of dependent and independent variables according to economic activity of observed companies are presented in Table 6.

**Table 6.** Descriptive statistics according to economic activity of companies.

| Activity | | N | Minimum | Maximum | Mean | Std. Deviation |
|---|---|---|---|---|---|---|
| Aquaculture Industry | | | | | | |
| | Growth | 31 | −0.8485 | 2.3023 | 0.1050 | 0.4934 |
| | Size | 31 | 3.5553 | 7.3499 | 5.1246 | 0.7866 |
| | Current Ratio | 31 | 0.2740 | 45.6670 | 4.5395 | 8.9899 |
| | ROA | 31 | −33.6890 | 40.2470 | 7.3114 | 14.4108 |
| | ROE | 31 | −73.5919 | 110.1396 | 16.7641 | 29.9712 |
| | Leverage | 31 | 0.0399 | 1.3778 | 0.4763 | 0.3079 |
| Breeding animal and production | | | | | | |
| | Growth | 628 | −2.4210 | 2.6852 | 0.0532 | 0.4759 |
| | Size | 628 | 0.6931 | 8.7328 | 5.0650 | 1.2122 |
| | Current Ratio | 628 | 0.1680 | 76.8140 | 4.1063 | 7.2110 |
| | ROA | 628 | −88.5530 | 99.9100 | 9.4510 | 16.2608 |
| | ROE | 628 | −225.5583 | 369.0842 | 18.4357 | 43.3164 |
| | Leverage | 628 | 0.0097 | 4.2228 | 0.4799 | 0.4331 |

**Table 6.** *Cont.*

| Activity | | N | Minimum | Maximum | Mean | Std. Deviation |
|---|---|---|---|---|---|---|
| Crop production | | | | | | |
| | Growth | 160 | −3.1092 | 2.5060 | 0.0047 | 0.5016 |
| | Size | 160 | 1.0986 | 8.4444 | 4.9074 | 1.2238 |
| | Current Ratio | 160 | 0.2070 | 75.5200 | 4.9720 | 10.0797 |
| | ROA | 160 | −54.2110 | 92.7370 | 10.0707 | 14.9175 |
| | ROE | 160 | −155.3201 | 320.3943 | 17.0816 | 39.2685 |
| | Leverage | 160 | 0.0088 | 3.8752 | 0.4981 | 0.4744 |
| Fruits and vegetable production | | | | | | |
| | Growth | 89 | −0.9879 | 2.0243 | 0.0347 | 0.4677 |
| | Size | 89 | 1.3863 | 7.1381 | 4.8679 | 1.2323 |
| | Current Ratio | 89 | 0.2380 | 77.7180 | 5.2409 | 10.4418 |
| | ROA | 89 | −83.5890 | 76.1380 | 8.4768 | 19.6053 |
| | ROE | 89 | −15.6524 | 274.9299 | 23.5644 | 38.9876 |
| | Leverage | 89 | 0.0084 | 4.2010 | 0.4969 | 0.5675 |
| Oilseed and Grain Farming Industry | | | | | | |
| | Growth | 321 | −2.3819 | 3.0797 | 0.1233 | 0.5367 |
| | Size | 321 | 1.0986 | 7.5224 | 5.1602 | 1.1137 |
| | Current Ratio | 321 | 0.1540 | 65.9260 | 4.4363 | 7.4267 |
| | ROA | 321 | −63.3250 | 89.2130 | 8.4159 | 14.5994 |
| | ROE | 321 | −81.0877 | 359.1302 | 19.5219 | 41.7386 |
| | Leverage | 321 | 0.0103 | 3.8874 | 0.4658 | 0.4352 |
| Mixed farming | | | | | | |
| | Growth | 41 | −1.0211 | 0.8644 | 0.0535 | 0.3249 |
| | Size | 41 | 2.4849 | 8.4185 | 5.3114 | 1.0589 |
| | Current Ratio | 41 | 0.1980 | 17.7290 | 2.9888 | 3.5715 |
| | ROA | 41 | −9.2360 | 65.4010 | 10.5290 | 15.9504 |
| | ROE | 41 | −87.8311 | 121.1862 | 17.7618 | 37.7889 |
| | Leverage | 41 | 0.0563 | 3.9765 | 0.5325 | 0.6308 |
| Silviculture and other forestry activities | | | | | | |
| | Growth | 63 | −1.4181 | 3.3715 | 0.0229 | 0.5411 |
| | Size | 63 | 2.4849 | 7.5380 | 5.3454 | 0.9106 |
| | Current Ratio | 63 | 0.2110 | 70.5810 | 4.8134 | 11.2177 |
| | ROA | 63 | −16.2720 | 36.1430 | 7.4969 | 10.8214 |
| | ROE | 63 | −135.1565 | 410.6464 | 13.1257 | 57.6836 |
| | Leverage | 63 | 0.0120 | 1.5499 | 0.4031 | 0.3085 |

Source: Author's calculation.

The highest average value of the sales growth rate had companies engaged in the cultivation of cereals, leguminous crops, and oil seeds in the observed period. On the other hand, crop production recorded the lowest average sales growth rates. The highest average value of company size had companies operating in the field of silviculture and other forestry activities with no significant value dispersions of this indicator. Judging by the average value of current ratio, we can conclude that companies in all observed activities of

sector A (agriculture, forestry, and fisheries) are characterized by liquid business. The most liquid business is recorded by companies in the field of fruits and vegetable production in observed period (5.2409). These companies have best aligned the deadlines for short-term assets bindings and maturities of short-term liabilities.

According to the reference value of ROA ($\geq$10%), we can conclude that only companies operating in crop production and mixed farming have achieved a satisfactory average rate of return on engaged assets in the amount of 10.0707% and 10.5290% respectively. Companies in other activities achieved a lower level of fertilization of engaging assets. It is interesting to note that companies in all activities have recorded a satisfactory average rate of return on engaged equity. The ability of engaged equity to result in the highest return has characterized the companies in the fruit and vegetable production (23.5644%). On the other hand, the lowest average return on employed equity was realized by companies operating in silviculture and other forestry activities (13.1257%) with significant value dispersion of this indicator, from $-135.1565$ to 410.6464. However, this is an activity in which companies have the most favorable financing structure since the total assets are 40.31% financed from debts. Judging by the average value of leverage, the highest percentage of debt-financed assets of companies was in the field of mixed farming, but the percentage of financing is within reasonable limits (53.25%).

The correlation analysis of the used variables is presented in Table 7. The growth of the company significantly correlated with three indicators. Positive correlation was noted with return on assets and leverage. Negative correlation was noted with the size of the company.

**Table 7.** Correlation matrix—Pearson correlation.

|  | Dependent | Size | Current Ratio | ROA | Leverage | ROE |
|---|---|---|---|---|---|---|
| Dependent | 1 | −0.108 | −0.010 | 0.128 | 0.039 | 0.081 |
| Size | −0.108 | 1 | 0.055 | 0.015 | −0.163 | −0.002 |
| Current Ratio | −0.010 | 0.055 | 1 | 0.235 | −0.328 | 0.021 |
| ROA | 0.128 | 0.015 | 0.235 | 1 | −0.318 | 0.441 |
| Leverage | 0.039 | −0.163 | −0.328 | −0.318 | 1 | 0.031 |
| ROE | 0.081 | −0.002 | 0.021 | 0.441 | 0.031 | 1 |

Source: Author's calculation.

Variance impact factors (VIF) for independent variables were shown in Table 8 in order to test multicollinearity. As can be seen in Table 8, VIF values were less than 5 for all variables. This assumption implies that there was no problem with multicollinearity [32]. Furthermore, Durbin-Watson value of 1.908 indicates that there is no autocorrelation (Table 9). Model summary in Table 9 indicates that there is no heteroskedasticity (Sig. F is less than 0.05).

**Table 8.** Variance impact factors of variables (VIF).

| Collinearity Statistics | |
|---|---|
| Tolerance | VIF |
| 0.971 | 1.029 |
| 0.872 | 1.147 |
| 0.679 | 1.474 |
| 0.782 | 1.278 |
| 0.771 | 1.297 |

Source: Author's calculation.

**Table 9.** Model summary.

| Model | R | R Square | Adjusted R Square | Change Statistics | | Durbin-Watson |
|---|---|---|---|---|---|---|
| | | | | R Square Change | Sig. F Change | |
| 1 | 0.603 | 0.340 | 0.305 | 0.340 | 0.000 | 1.908 |

Source: Author's calculation.

The presented results in Table 10 showed that the proposed model was statistically significant with $p < 0.05$. The first variable, company size, had a statistically significant negative impact on the growth of observed companies ($p < 0.05$) which means that hypothesis 1 is rejected. Gibrat's assumption that firm size is not relevant for growth is also rejected. A smaller size leads to a greater opportunity for the company to grow, with the growth rate decreasing as the size of the company increases. Therefore, large companies in the agriculture, forestry, and fisheries sector have either already reached the optimal size or are close to it, which causes a very small growth of these companies or even the need to reduce their size if the optimal size is exceeded. Therefore, the observed companies are looking for the optimal, most efficient size, given that the advantages of efficiency are related to economies of scale. There is a negative relationship between the size and growth of the observed companies until the optimal size is achieved. After reaching the optimal size, the size of the company begins to have a positive impact on the growth of the company. Bearing in mind that the size of the observed companies is measured by the logarithm of the number of employees, and that the growth of the number of employees is inversely related to the growth of the company, this may indicate low productivity or low technical equipment of employees. The inverse relationship was in accordance with the research conducted by Voulgaris et al. [3] who found that small companies usually achieve higher growth rates given that growth is achieved on a lower basis and because it is necessary to achieve efficient size as soon as possible. The negative relationship was also in accordance with the research of Simbana et al. whose results showed that larger Ecuadorian companies from all economic sectors have lower growth than smaller ones [28]. By using quantile regressions on a sample of 2278 Portuguese small and medium companies for the time period of 1999 to 2006, Serrasqueiro et al. also confirmed the negative relationship between size and growth in circumstances when the size of these companies is growing considerably [33].

**Table 10.** Multiple regression model.

| Model | Unstandardized Coefficients | | Standardized Coefficients | t | Sig. |
|---|---|---|---|---|---|
| | B | Std. Error | Beta | | |
| (Constant) | 0.680 | 0.201 | | 3.386 | 0.001 |
| Size | −0.126 | 0.035 | −0.099 | −3.625 | 0.000 |
| Current Ratio | −0.003 | 0.005 | −0.019 | −0.656 | 0.512 |
| ROA | 0.014 | 0.003 | 0.148 | 4.515 | 0.000 |
| Leverage | 0.211 | 0.101 | 0.064 | 2.085 | 0.037 |
| ROE | 0.001 | 0.001 | 0.014 | 0.469 | 0.639 |

Source: Author's calculation.

The variable profitability measured by ROA had a statistically significant positive impact on the growth of observed companies ($p < 0.05$). So, hypothesis 3 is confirmed. The realized profit is an important precondition for future company growth. Companies in this sector strive to maximize profits by achieving a return on engaged funds and to preserve the real value of invested net assets, which has a positive effect on the growth of these companies. The profitability and growth are the basic indicators of the company business success set by the management. The process of continuously improving profitability is the company growth. Observed large and very large companies make high profits since

they have a dominant position in the industry and achieve higher sales growth rates. The high profitability of these companies may be the result of low input prices and favorable lending conditions. The assumption of sustainable growth of these companies is achieved profitability. The positive relationship between profitability and growth was also confirmed in research conducted by Kachlami and Yazdanfar [10] who start from the fact that more profitable companies use external sources of financing to a greater extent and achieve higher growth. Megaravalli and Sampagnaro [31] also confirmed that an increase in return on assets leads to an increase in the probability that the company will achieve higher growth. The obtained results indicate that the profitability of companies in the sector of agriculture, forestry, and fisheries had a positive impact on growth is in line with the results of research of U.S. restaurant companies whose profitability in the previous year had a positive impact on current year growth, which implies that profit creates growth [34]. On the other hand, the variable profitability measured by ROE also has a positive, but not a statistically significant, impact on the growth of observed companies ($p > 0.05$). So, hypothesis 4 is rejected. Therefore, a higher rate of return on the engaged capital by companies of agriculture, forestry, and fisheries sectors does not affect the growth of these companies. This result is in accordance with the result of the research conducted by Markman and Gartner [35] who also confirmed that there is no significant relationship between profitability and growth, measured in absolute or relative value.

The variable leverage had a statistically significant positive impact on the growth of observed companies ($p < 0.05$). So, hypothesis 5 is confirmed. As one of the most important indicators of solvency and financial security, leverage represents the increased volume of financing from other sources that are sustainable in the long run only if observed companies achieve a higher rate of return on total equity than the price they pay for the use of other sources of capital. The seasonal nature of the production of agriculture companies affects the slow turnover of capital. Due to the insufficient volume of their accumulation, companies in this sector use external sources of financing in order to create a more competitive and productive agricultural production. It is noticeable that companies in this sector use borrowed sources of financing on a large scale to finance their assets and there is a strong ability of these companies to achieve growth and development through additional debts. However, there is no problem with their over-indebtedness, given that the increased volume of financing from other sources has a positive effect on the growth of these companies. The positive relationship was also confirmed in research conducted by Heshmati [36]; Hermelo and Vassolo [15]; and Huynh and Petrunia [37]. Becchetti and Trovato [14] found that companies with a higher level of leverage are growing much faster than companies that are financed to a much lesser extent from their sources of financing. Hameed et al. [38] also confirmed the positive relationship between leverage and company growth of non-financial companies selected from Karachi Stock Exchange, suggesting that companies with high levels of leverage should reduce their share of debt for increasing their assets and maintain market growth. According to Serrasqueiro et al., leverage is an indicator that encourages growth in the case of a large increase in the size of small and medium companies [33].

The variable current liquidity has also a positive, but not statistically significant, impact on the growth of observed companies ($p > 0.05$). So, hypothesis 2 is rejected. Achieving optimal liquidity is one of the key concepts from the aspect of survival, sustainable growth, and development of the company. Achieving an optimal level of company liquidity is important, as a too high level of liquidity indicates a surplus of cash funds that are not for investment purposes and do not result in future economic benefits [39]. Liquid business of European companies in the agriculture, forestry, and fisheries sectors means that they have managed to adjust the time of settling short-term liabilities and collect receivables in order to supply the business cycle with basic inputs on time. Therefore, the more liquid business of companies in the agriculture, forestry, and fisheries sectors has a positive effect on growth, but this effect is not statistically significant. The obtained results are in accordance with the empirical study conducted by Khaldun and Muda [21] who researched liquidity

ratios that influence profit growth. The results showed that the liquidity ratios together significantly influence the growth of profit of manufacturing companies in the food and beverages sector listed on the Indonesia Stock Exchange in the time from 2010 to 2012. The impact of current liquidity on profit growth is positive, but not statistically significant. In addition, conducting research on a sample of Italian manufacturing companies for the time period of 1995 to 2000, Fagiolo and Luzzi noted that liquidity constraints should also be taken into account as stronger liquidity constraints lead to lower growth rates in the future [40].

## 5. Conclusions

Company growth in competitive business conditions affects employment growth, general social well-being, and economic development. In the process of growth, companies are trained to continuously increase the effectiveness and efficiency of operations as well as to achieve increased profit in the function of profit maximization as a long-term business goal. The primary goal of this research was to determine the main indicators of company growth in the agriculture, forestry, and fisheries sector based on data from financial statements during the six year period between 2014 and 2019. The paper has analyzed the effect of five independent variables on the growth of 1333 observations of companies in the European large and very large companies. The independent variables observed were company size, current ratio, return on total assets, return on equity, and leverage. The growth measured by the sales growth rate was considered as a dependent variable. The multiple regression analysis was applied in order to research the key variables that have a determining impact on the growth of the observed companies. Obtained empirical evidence showed that smaller European companies in the agriculture, forestry, and fisheries sector grow faster than larger ones, rejecting the assumption of Gibrat's Law. Additionally, these companies with a higher rate of return on engaged assets achieve higher growth rates. The observed companies rely heavily on the use of external sources to finance their activities in order to achieve a higher growth rate.

This research provides a better understanding of improvement geared towards greater growth of companies in the agriculture, forestry, and fisheries field, taking into account variables such as company size, profitability, and leverage. The results of this research can serve all external stakeholders such as managers, owners, employees, and shareholders. Knowing the determinants that affect the growth of companies, managers of companies in the agriculture, forestry, and fisheries sector will be able to influence the achievement of other company goals through greater growth, such as increasing market share, reducing operating costs, increasing profits, and reducing the impact of competing companies. This analysis can be useful for investors to determine which indicators affect the company growth in order to define the amount of future investments. The knowledge of the company growth rate level is important in the context of achieving competitive advantage through lower production costs in relation to economies of scale and the scope and degree of business risk diversification. The comparative advantages of this sector that companies should use are reflected in the richness and quality of natural resources, as well as the good geographical position. The intensive development of this sector in the future could be realized at the expense of the intensive development of the industry through measures and incentives determined by economic policymakers.

Improving the business of companies in the sector of agriculture, forestry, and fisheries is aimed at more efficient use of economic resources, technical and technological development, management of natural resources, and protection of the natural environment. It is therefore necessary to ensure high revenues and growth of these companies, without destroying natural resources. In order to foster sustainability, companies in the agriculture, forestry, and fisheries sector should ensure the strengthening of production and competition in this sector, as well as promote the economic opportunities of agricultural production. In that way, it will satisfy human needs for food in the long run, preserve the quality of the environment and natural resources, provide high economic value, and improve the

quality of society as a whole. Sustainable business of companies in this sector is aimed at ensuring profitable business, healthy environment, economic profitability, and economic and social justice.

The limitations of this research represent recommendations for future research. First of all, the research is limited to the territory of Europe and companies operating in the agriculture, forestry, and fisheries sectors. Further research may cover companies operating in other geographical areas and other industries. Since the analysis was done based on financial statements, some other financial determinants that affect growth could be investigated in future research. In addition, non-financial determinants that affect growth could be included. The study is limited to the period from 2014 to 2019. Analysis of a longer period of time would certainly give more precise results. However, the conducted empirical study surely represents the basis for research in the coming period that will analyze the type of the relationship between key variables and company growth. Further research could be also aimed at determining the level of sustainable growth that the observed companies can achieve with the given financial resources. The growth of a company at a higher rate than the sustainable growth rate leads to financial problems, i.e., bankruptcy in the final instance. On the other hand, the growth of a company at a lower rate than the sustainable growth rate leads to the stagnation of the company. The identification of a sustainable growth rate as a financial planning tool is important in the context of assessing past growth results and for setting guidelines for future company growth as well.

**Author Contributions:** Conceptualization, V.M. and D.J.; methodology, K.P.; investigation, B.V.; resources, S.M.; data curation, K.P.; writing—original draft preparation, B.V. and S.M.; visualization, B.V.; supervision, V.M. and D.J. All authors have read and agreed to the published version of the manuscript.

**Funding:** This research received no external funding.

**Institutional Review Board Statement:** Not applicable.

**Informed Consent Statement:** Not applicable.

**Data Availability Statement:** Not applicable.

**Conflicts of Interest:** The authors declare no conflict of interest. The funders had no role in the design of the study; in the collection, analyses, or interpretation of data; in the writing of the manuscript or in the decision to publish the results.

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
