# Peer review of "The Analysis of Company Growth Determinants Based on Financial Statements of the European Companies"

_sustainability, doi:10.3390/su14020770_

Round 1
Reviewer 1 Report
#1 The scope of "Determinants of Company Growth" is too wide, and whether the author's research content can represent the scope of this topic is open to question.
#2 What are the characteristics of the author’s research hypothesis and regression, which can represent the practical value of the "Determinants of Company Growth: Evidence from the European Companies" research. It is recommended that the author should provide more rigorous research supporting materials, so that the research topics are similar to those of other past scholars. There is a clear distinction, and it is recommended that the author should narrow the source basis of the sample. If the entire European company is used as the sample, the authors need to explore the differences between different countries in addition to the overall European market.
#3 With the current research content, it is impossible to provide a rigorous academic process.At the same time, whether the selection of samples for consideration by European companies is representative is also debatable.
Reviewer 2 Report
The article is well written and data-based. Bringing the Results and Discussions together into one point may be disturbing, but it would not be advisable to separate these two parts in this case. In my opinion, the Conclusions are correctly written and relate to both scientific and practical issues. One could consider deepening the literature research, but it does not affect my positive assessment of this text.
Author Response
Thank you for your comments. The authors have added more sources of literature.
Reviewer 3 Report
The paper is focused on the company growth in the field of agriculture, forestry and fishing. The aim of this paper is to figured out main factors affecting this growth.
The authors analyze the financial and economic results of a sample or 268 companies in Europe. The methodology is a linear regression model where the dependent variable is the growth while independents variables are Size, Liquity, ROA, ROE, leverage.
The model shows significant effects on company growth for Size, ROA and Leverage.
The paper is well structured and research objectives are quite clear. The literature is extended. Conclusions are explained too.
THE SAMPLE, THE MODEL AND RESULTS SHOW SOME CRITICAL POINTS
First, the company size is defined in the Table 1 as the number of employees. When reading the paper it seems that the company size be the turnover (value of the sales).
The model is basic. The results are not original.
ABOUT THE THEORY
The authors evaluate factors affecting the economic growth. Looking at the model, I can see as exogenous factors the number of employees, the net income (which actually depend on sales), the value of asset as total and equity. I would suggest authors to introduce the investments if they are available or liabilities (as loans, not short term debts) which depend on the company decisions about investments.
Still, the authors estimate just economic or financial performance as growth, employees, ROE or ROA. However, all these variables are affected by the behavior of the company such the structure–conduct–performance (SCP) paradigm (Chamberlin and Robinson, 1933). In this context, the change in the market share of the company may be a better indicator as well as the size of the market or the number of importing countries.
If the sample is made by farms then the authors should pay attention to the product supplied avoiding the aggregation among agriculture, forestry and fishery. The economic performance of the farm can be hardly assessed through ROA or ROE. The authors should explain this point.
ABOUT THE SAMPLE
The authors analyze the financial and economic results of a sample or 268 companies in Europe.
Nothing is said about this sample. The authors should report some descriptive stats for the sample. For instance, are these companies farms or processors? or where are these companies located (in which countries) and especially what they are doing? Are they working in the dairy sector, in wine? in the pasta? Usually, results as profitability differ strongly from a production to another.
Are these companies or corporate? So are they producing homogenous products or different products
However, I think that 268 is really small number for a European sample. Looking at the size I may suppose that these are farms (5 employees as average). The authors should better explain this point.
Table 2 – desc stat. I see very high numbers for ROA and ROE (for instance the max ROE is 410!). This is a very high number respect to my experience, at least in my country where food industry is very important.
The authors should explain these results.
ABOUT THE MODEL
The Growth is the change of company turnover year after year. An increase in sales usually leads to an increase in the gross or net income even if the impact of investments may affect the ROA.
The ROE (net income / equity) differs from the ROA (net income / total assets) because of the asset financed by liabilities.
I do not understand why ROI has not used. Unless big corporates working in different activities the total asset is usually the asset functional for the company.
It seems that this model suffers some endogeneity. For instance bot ROA and ROE are actually a consequence of the net income and asset.
Authors did not test the heteroskedasticity which is an assumption in the linear regression algorithm.
ABOUT RESULTS
The authors should reflect on some statements. For instance, the statement “The growth is the basis for providing competitive advantages that are long-term sustainable and profitable” (page 9 row 371) is not correct. The growth is just a mathematical result of the increase in the sales, i.e., the effect not the cause. The competitive advantage should be related to market share, to prices, to product quality, to the attractiveness of company products in other countries, etc.
The growth of sales being a change is usually high when the sales are small and low when the sales are high. This is just a statistical effect. However, big companies produce high turnovers because they sale high volumes at lower prices or the sale few high quality products or they are able to produce excellent products (that have high selling prices).
The English is good. I suggest authors to use the present tense when possible.
Reviewer 4 Report
The study appears well structured, the research moves in a correct and functional way, the paragraphs are clear, exhaustive and well coordinated.
What appears useful to ask is whether the five indicators considered can be fully expressive also for the primary sector. It should be underlined, in fact, that a correct analysis of agricultural, forestry and fishing companies must be placed in the place of activity. The primary activity is affected more than others by the location for which parameters that cannot be proposed in the plains become acceptable or typical in the mountains or in marginal conditions. Furthermore, the forestry sector has strong peculiarities: it is a sector that requires very high anticipation capital and is characterized by payback times that can even exceed two decades. Despite this, I find interesting the idea developed in the work of developing an analysis methodology in the primary sector, and in the three sectors (agriculture, forestry and fisheries), characterized by particularly high risks and production costs and lower capital investments . Another element to consider is having considered medium / large farms, while we know that the sustainability of the agricultural sector, fisheries and, albeit to a lesser extent, also the forestry sector, passes through the multiplicity of small and micro enterprises that support entire territories and smaller productions but of great importance in terms of biodiversity and economic support. The originality of the work remains valid, which moves in the sense of research rather than of the determinants (of certain determinants) that lead to the growth of these companies, and which translate into a push towards the improvement of knowledge with a view to developing the sector. It is advisable to consider these two elements also in the development of the work.
Round 2
Reviewer 1 Report
#1 The title of "Determinants of Company Growth: Evidence from the European Companies" is too broad, and the information collected by the author cannot cover all the content related to "Determinants of Company Growth".
#2 The author did not systematically address the content of the defense, how to cite appropriate references to support the research meaning and value of the research topic. It is recommended that the author should state in detail how to modify the content of the defense, so that the reviewer can judge whether the modified content is appropriate.
#3 Like the response in #2, the author should reply to the revised content in detail, and at the same time systematically introduce the development process of research methods and analysis materials, based on this, readers can know what the author is talking about and the rigorous and practical judgment basis of the relevant content .
Reviewer 3 Report
Sustainability Journal
Title: Determinants of Company Growth: Evidence from the European Companies
2 revision
The paper is focused on the company growth in the field of agriculture, forestry and fishing. The aim of this paper is to figured out main factors affecting this growth.
I appreciate the authors effort in revising the paper.
Still the changes does not enrich significantly the paper.
The authors replied to past comments. Some replies correspond to paper changes, others not at all.
The sample is better defined but there are still unclear points.
The main one is about the sample units. The authors reply as “ large and very large companies categorized as public limited companies, private limited companies, limited liability companies, joint-stock companies and private joint-stock companies” and “code of activity and included companies operating in the sector of agriculture, forestry and fisheries on the European market, and they are producing different products. There is no farms”. Which code? NACE/ISIC? Is Tp Catalyst database in line with international economic activity classifications? I suppose that these “companies” are working on manufacturing (food and/or beverages) or on trading (agricultural, food products). The code A divisions 1,2,3 is actually about “Agriculture, forestry and fishing” that includes farming, silviculture, fishing or aquaculture. In this sectors holdings are usually small and mostly do not have a national mandatory accounting systems.
According to this, Table 2 seems not relevant
Another point is about the country of these companies. Why Eastern or small countries are chosen? Why the authors do not chose countries such as Germany, France or Great Britain or Italy or Spain where the food sector is much more representative of the World food business?
The reply to my comment “Table 2 – desc stat. I see very high numbers for ROA and ROE (for instance the max ROE is 410!). This is a very high number respect to my experience, at least in my country where food industry is very important. The authors should explain these results” .is not satisfactorily and trivial.
The results as ROA and ROE depend strongly on economic activities carried out by sample units. Descriptive stats should be rearranged by economic activity (eg. Meat or vegetables) or by economic size (small vs. big). The aggregation is a strong limit in understanding profitability.
Round 3
Reviewer 1 Report
#1 Such a title is still too broad, and it is more appropriate for authors to limit the scope of the research to the discussion of the paper.
#2 Well done than the previous version.
#3 Well done than the previous version.
Author Response
,, Please see the attachment"
